# The committee machine: Computational to statistical gaps in learning a two-layers neural network

**Benjamin Aubin**[*†], **Antoine Maillard**[†], **Jean Barbier**[⊗◇†]
**Florent Krzakala**[†], **Nicolas Macris**[⊗], **Lenka Zdeborová**[*]

## Abstract

Heuristic tools from statistical physics have been used in the past to locate the phase transitions and compute the optimal learning and generalization errors in the teacher-student scenario in multi-layer neural networks. In this contribution, we provide a rigorous justification of these approaches for a two-layers neural network model called the committee machine. We also introduce a version of the approximate message passing (AMP) algorithm for the committee machine that allows to perform optimal learning in polynomial time for a large set of parameters. We find that there are regimes in which a low generalization error is information-theoretically achievable while the AMP algorithm fails to deliver it; strongly suggesting that no efficient algorithm exists for those cases, and unveiling a large computational gap.

While the traditional approach to learning and generalization follows the Vapnik-Chervonenkis [1] and Rademacher [2] worst-case type bounds, there has been a considerable body of theoretical work on calculating the generalization ability of neural networks for data arising from a probabilistic model within the framework of statistical mechanics [3, 4, 5, 6, 7]. In the wake of the need to understand the effectiveness of neural networks and also the limitations of the classical approaches [8], it is of interest to revisit the results that have emerged thanks to the physics perspective. This direction is currently experiencing a strong revival, see e.g. [9, 10, 11, 12].

Of particular interest is the so-called teacher-student approach, where labels are generated by feeding i.i.d. random samples to a neural network architecture (the *teacher*) and are then presented to another neural network (the *student*) that is trained using these data. Early studies computed the information theoretic limitations of the supervised learning abilities of the teacher weights by a student who is given $m$ independent $n$-dimensional examples with $\alpha \equiv m/n = \Theta(1)$ and $n \to \infty$ [3, 4, 7]. These works relied on non-rigorous heuristic approaches, such as the replica and cavity methods [13, 14]. Additionally no provably efficient algorithm was provided to achieve the predicted learning abilities, and it was thus difficult to test those predictions, or to assess the computational difficulty.

Recent developments in statistical estimation and information theory —in particular of approximate message passing algorithms (AMP) [15, 16, 17, 18], and a rigorous proof of the replica formula for the optimal generalization error [11]— allowed to settle these two missing points for single-layer neural networks (i.e. without any hidden variables). In the present paper, we leverage on these works, and provide rigorous asymptotic predictions and corresponding message passing algorithm for a class of two-layers networks.

[*] Institut de Physique Théorique, CNRS & CEA & Université Paris-Saclay, Saclay, France.
[†] Laboratoire de Physique Statistique, CNRS & Sorbonnes Universités & École Normale Supérieure, PSL University, Paris, France.
[⊗] Laboratoire de Théorie des Communications, Faculté Informatique et Communications, Ecole Polytechnique Fédérale de Lausanne, Suisse.
[◇] International Center for Theoretical Physics, Trieste, Italy.

# 1 Summary of contributions and related works

While our results hold for a rather large class of non-linear activation functions, we illustrate our findings on a case considered most commonly in the early literature: The committee machine. This is possibly the simplest version of a two-layers neural network where all the weights in the second layer are fixed to unity. Denoting $Y_\mu$ the label associated with a $n$-dimensional sample $X_\mu$, and $W_{il}^*$ the weight connecting the $i$-th coordinate of the input to the $l$-th node of the hidden layer, it is defined by:

$$Y_\mu = \text{sign}\Big[ \sum_{l=1}^{K} \text{sign}\Big( \sum_{i=1}^{n} X_{\mu i} W_{il}^* \Big) \Big]. \tag{1}$$

We concentrate here on the teacher-student scenario: The teacher generates i.i.d. data samples with i.i.d. standard Gaussian coordinates $X_{\mu i} \sim \mathcal{N}(0,1)$, then she/he generates the associated labels $Y_\mu$ using a committee machine as in (1), with i.i.d. weights $W_{il}^*$ unknown to the student (in the proof section we will consider the more general case of a distribution for the weights of the form $\prod_{i=1}^{n} P_0(\{W_{il}^*\}_{l=1}^{K})$, but in practice we consider the fully separable case). The student is then given the $m$ input-output pairs $(X_\mu, Y_\mu)_{\mu=1}^{m}$ and knows the distribution $P_0$ used to generate $W_{il}^*$. The goal of the student is to learn the weights $W_{il}^*$ from the available examples $(X_\mu, Y_\mu)_{\mu=1}^{m}$ in order to reach the smallest possible generalization error (i.e. to be able to predict the label the teacher would generate for a new sample not present in the training set).

There have been several studies of this model within the non-rigorous statistical physics approach in the limit where $\alpha \equiv m/n = \Theta(1)$, $K = \Theta(1)$ and $n \to \infty$ [19, 20, 21, 22, 6, 7]. A particularly interesting result in the teacher-student setting is the *specialization of hidden neurons* (see sec. 12.6 of [7], or [23] in the context of online learning): For $\alpha < \alpha_{\text{spec}}$ (where $\alpha_{\text{spec}}$ is a certain critical value of the sample complexity), the permutational symmetry between hidden neurons remains conserved even after an optimal learning, and the learned weights of each of the hidden neurons are identical. For $\alpha > \alpha_{\text{spec}}$, however, this symmetry gets broken as each of the hidden units correlates strongly with one of the hidden units of the teacher. Another remarkable result is the calculation of the optimal generalization error as a function of $\alpha$.

Our first contribution consists in a proof of the replica formula conjectured in the statistical physics literature, using the adaptive interpolation method of [24, 11], that allows to put several of these results on a firm rigorous basis. Our second contribution is the design of an AMP-type of algorithm that is able to achieve the optimal learning error in the above limit of large dimensions for a wide range of parameters. The study of AMP —that is widely believed to be optimal between all polynomial algorithms in the above setting [25, 26, 27, 28]— unveils, in the case of the committee machine with a large number of hidden neurons, the existence a large *hard phase* in which learning is information-theoretically possible, leading to a good generalization error decaying asymptotically as $1.25K/\alpha$ (in the $\alpha = \Theta(K)$ regime), but where AMP fails and provides only a poor generalization that does not decay when increasing $\alpha$. This strongly suggests that no efficient algorithm exists in this hard region and therefore there is a computational gap in learning in such neural networks. In other problems where a hard phase was identified, its study boosted the development of algorithms that are able to match the predicted thresholds and we anticipate this will translate to the present model.

We also want to comment on a related line of work that studies the loss-function landscape of neural networks. While a range of works show under various assumptions that spurious local minima are absent in neural networks, others show under different conditions that they do exist, see e.g. [29]. The regime of parameters that is hard for AMP must have spurious local minima, but the converse is not true in general. It might be that there are spurious local minima, yet the AMP approach succeeds. Moreover, in all previously studied models in the Bayes-optimal setting the (generalization) error obtained with the AMP is the best known and other approaches, e.g. (noisy) gradient based, spectral algorithms or semidefinite programming, are not better in generalizing even in cases where the "student" models are overparametrized. Of course in order to be in the Bayes-optimal setting one needs to know the model used by the teacher which is not the case in practice.

# 2 Main technical results

**A general model —** While in the illustration of our results we shall focus on the model (1), all our formulas are valid for a broader class of models: Given $m$ input samples $(X_{\mu i})_{\mu, i=1}^{m,n}$, we denote $W_{il}^*$

the teacher-weight connecting the $i$-th input (i.e. visible unit) to the $l$-th node of the hidden layer. For a generic function $\varphi_{\text{out}} : \mathbb{R}^K \times \mathbb{R} \to \mathbb{R}$ one can formally write the output as

$$Y_\mu = \varphi_{\text{out}}\Big(\Big\{\frac{1}{\sqrt{n}}\sum_{i=1}^{n} X_{\mu i} W_{il}^*\Big\}_{l=1}^{K}, A_\mu\Big) \quad \text{or} \quad Y_\mu \sim P_{\text{out}}\Big(\cdot \Big| \Big\{\frac{1}{\sqrt{n}}\sum_{i=1}^{n} X_{\mu i} W_{il}^*\Big\}_{l=1}^{K}\Big), \quad (2)$$

where $(A_\mu)_{\mu=1}^{m}$ are i.i.d. real valued random variables with known distribution $P_A$, that form the probabilistic part of the model, generally accounting for noise. For deterministic models the second argument is simply absent (or is a Dirac mass). We can view alternatively (2) as a channel where the transition kernel $P_{\text{out}}$ is directly related to $\varphi_{\text{out}}$. As discussed above, we focus on the teacher-student scenario where the teacher generates Gaussian i.i.d. data $X_{\mu i} \sim \mathcal{N}(0,1)$, and i.i.d. weights $W_{il}^* \sim P_0$. The student then learns $W^*$ from the data $(X_\mu, Y_\mu)_{\mu=1}^{m}$ by computing marginal means of the posterior probability distribution (5).

Different scenarii fit into this general framework. Among those, the committee machine is obtained when choosing $\varphi_{\text{out}}(h) = \text{sign}(\sum_{l=1}^{K} \text{sign}(h_l))$ while another model considered previously is given by the parity machine, when $\varphi_{\text{out}}(h) = \prod_{l=1}^{K} \text{sign}(h_l)$, see e.g. [7]. A number of layers beyond two has also been considered, see [22]. Other activation functions can be used, and many more problems can be described, e.g. compressed pooling [30, 31] or multi-vector compressed sensing [32].

**Two auxiliary inference problems —** Denote $\mathcal{S}_K$ the finite dimensional vector space of $K \times K$ matrices, $\mathcal{S}_K^+$ the convex and compact set of semi-definite positive $K \times K$ matrices, $\mathcal{S}_K^{++}$ for positive definite $K \times K$ matrices, and $\forall N \in \mathcal{S}_K^+$ we set $S_K^+(N) \equiv \{M \in S_K^+ \text{ s.t. } N - M \in \mathcal{S}_K^+\}$.

Stating our results requires introducing two simpler auxiliary $K$-dimensional estimation problems:

• The first one consists in retrieving a $K$-dimensional input vector $W_0 \sim P_0$ from the output of a Gaussian vector channel with $K$-dimensional observations $Y_0 = r^{1/2} W_0 + Z_0$, $Z_0 \sim \mathcal{N}(0, I_{K \times K})$ and the "channel gain" matrix $r \in \mathcal{S}_K^+$. The associated posterior distribution on $w = (w_l)_{l=1}^{K}$ is

$$P(w|Y_0) = \frac{1}{\mathcal{Z}_{P_0}} P_0(w) e^{Y_0^\intercal r^{1/2} w - \frac{1}{2} w^\intercal r w}, \quad (3)$$

and the associated *free entropy* (or minus *free energy*) is given by the expectation over $Y_0$ of the log-partition function $\psi_{P_0}(r) \equiv \mathbb{E} \ln \mathcal{Z}_{P_0}$ and involves $K$ dimensional integrals.

• The second problem considers $K$-dimensional i.i.d. vectors $V, U^* \sim \mathcal{N}(0, I_{K \times K})$ where $V$ is considered to be known and one has to retrieve $U^*$ from a scalar observation obtained as $\widetilde{Y}_0 \sim P_{\text{out}}(\cdot|q^{1/2}V + (\rho - q)^{1/2}U^*)$ where the second moment matrix $\rho \equiv \mathbb{E}[W_0 W_0^\intercal]$ is in $\mathcal{S}_K^+$ ($W_0 \sim P_0$) and the so-called "overlap matrix" $q$ is in $S_K^+(\rho)$. The associated posterior is

$$P(u|\widetilde{Y}_0, V) = \frac{1}{\mathcal{Z}_{P_{\text{out}}}} \frac{e^{-\frac{1}{2}u^\intercal u}}{(2\pi)^{K/2}} P_{\text{out}}\big(\widetilde{Y}_0|q^{1/2}V + (\rho - q)^{1/2}u\big), \quad (4)$$

and the free entropy reads this time $\Psi_{P_{\text{out}}}(q; \rho) \equiv \mathbb{E} \ln \mathcal{Z}_{P_{\text{out}}}$ (with the expectation over $\widetilde{Y}_0$ and $V$) and also involves $K$ dimensional integrals.

**The free entropy —** The central object of study leading to the optimal learning and generalization errors in the present setting is the posterior distribution of the weights:

$$P(\{w_{il}\}_{i,l=1}^{n,K} | \{X_{\mu i}, Y_\mu\}_{\mu,i=1}^{m,n}) = \frac{1}{\mathcal{Z}_n} \prod_{i=1}^{n} P_0(\{w_{il}\}_{l=1}^{K}) \prod_{\mu=1}^{m} P_{\text{out}}\Big(Y_\mu \Big| \Big\{\frac{1}{\sqrt{n}}\sum_{i=1}^{n} X_{\mu i} w_{il}\Big\}_{l=1}^{K}\Big), \quad (5)$$

where the normalization factor is nothing else than a *partition function*, i.e. the integral of the numerator over $\{w_{il}\}_{i,l=1}^{n,K}$. The expected [2] free entropy is by definition $f_n \equiv \mathbb{E} \ln \mathcal{Z}_n / n$. The replica formula gives an explicit (conjectural) expression of $f_n$ in the high-dimensional limit $n, m \to \infty$ with $\alpha = m/n$ fixed. We discuss in the long version of this paper [33] how the heuristic replica method [13, 14] yields the formula. This computation was first performed, to the best of our knowledge, by [19] in the case of the committee machine. Our first contribution is a rigorous proof of the corresponding free entropy formula using an interpolation method [34, 35, 24].

In order to formulate our rigorous results, we add an (arbitrarily small) Gaussian regularization noise $Z_\mu \sqrt{\Delta}$ to the first expression of the model (2), where $\Delta > 0$, $Z_\mu \sim \mathcal{N}(0,1)$, so that the channel kernel is ($u \in \mathbb{R}^K$)

$$P_{\text{out}}(y|u) = \frac{1}{\sqrt{2\pi\Delta}} \int_{\mathbb{R}} \mathrm{d}P_A(a) e^{-\frac{1}{2\Delta}(y-\varphi(u,a))^2}. \tag{6}$$

**Theorem 2.1 (Replica formula)** *Suppose (H1): The prior $P_0$ has bounded support in $\mathbb{R}^K$; (H2): The activation $\varphi_{\text{out}} : \mathbb{R}^K \times \mathbb{R} \to \mathbb{R}$ is a bounded $\mathcal{C}^2$ function with bounded first and second derivatives w.r.t. its first argument (in $\mathbb{R}^K$-space); and (H3): For all $\mu = 1, \ldots, m$ and $i = 1, \ldots, n$ we have i.i.d. $X_{\mu i} \sim \mathcal{N}(0,1)$. Then for the model (2) with kernel (6) the limit of the free entropy is:*

$$\lim_{n\to\infty} f_n \equiv \lim_{n\to\infty} \frac{1}{n} \mathbb{E} \ln \mathcal{Z}_n = \sup_{r \in \mathcal{S}_K^+} \inf_{q \in \mathcal{S}_K^+(\rho)} \left\{ \psi_{P_0}(r) + \alpha \Psi_{P_{\text{out}}}(q;\rho) - \frac{1}{2}\text{Tr}(rq) \right\}, \tag{7}$$

*where $\alpha \equiv m/n$ and where $\Psi_{P_{\text{out}}}(q;\rho)$ and $\psi_{P_0}(r)$ are the free entropies of two simpler $K$-dimensional estimation problems (3) and (4).*

This theorem extends the recent progress for generalized linear models of [11], which includes the case $K = 1$ of the present contribution, to the phenomenologically richer case of two-layers problems such as the committee machine. The proof sketch based on an *adaptive interpolation method* recently developed in [24] is outlined and the details can be found in [33]. Note that, following similar approximation arguments as in [11], the hypothesis (H1) can be relaxed to the existence of the second moment of the prior; thus covering the Gaussian case, (H2) can be dropped (and thus include model (1) and its $\text{sign}(\cdot)$ activation) and (H3) extended to data matrices $X$ with i.i.d. entries of zero mean, unit variance and finite third moment. Moreover, the case $\Delta = 0$ can be considered when the outputs are discrete, as in the committee machine (1), see [11]. The channel kernel becomes $P_{\text{out}}(y|u) = \int \mathrm{d}P_A(a) \mathbf{1}(y - \varphi(u,a))$ and the replica formula is the limit $\Delta \to 0$ of the one provided in Theorem 2.1. In general this regularizing noise is needed for the free entropy limit to exist.

**Learning the teacher weights and optimal generalization error —** A classical result in Bayesian estimation is that the estimator $\hat{W}$ that minimizes the mean-square error with the ground-truth $W^*$ is given by the expected mean of the posterior distribution. Denoting $q^*$ the extremizer in the replica formula (7), we expect from the replica method that in the limit $n \to \infty$, $m/n = \alpha$, that with high probablity $\hat{W}^\intercal W^*/n \to q^*$. We refer to proposition 4.2 and to the proof in [33] for the precise statement, that remains rigorously valid *only* in the presence of an additional (possibly infinitesimal) side-information. From the overlap matrix $q^*$, one can compute the Bayes-optimal generalization error when the student tries to classify a new, yet unseen, sample $X_{\text{new}}$. The estimator of the new label $\hat{Y}_{\text{new}}$ that minimizes the mean-square error with the true label is given by computing the posterior mean of $\varphi_{\text{out}}(X_{\text{new}}w)$ ($X_{\text{new}}$ is a row vector). Given the new sample, the optimal generalization error is then

$$\frac{1}{2}\mathbb{E}\left[ \left( \mathbb{E}_{w|X,Y}[\varphi_{\text{out}}(X_{\text{new}}w)] - \varphi_{\text{out}}(X_{\text{new}}W^*) \right)^2 \right] \xrightarrow[n\to\infty]{} \epsilon_g(q^*) \tag{8}$$

where $w$ is distributed according to the posterior measure (5) (note that this Bayes-optimal computation differs from the so-called Gibbs estimator by a factor 2, see [33]). In particular, when the data $X$ is drawn from the standard Gaussian distribution on $\mathbb{R}^{m\times n}$, and is thus rotationally invariant, it follows that this error only depends on $w^\intercal W^*$, which converges to $q^*$. Then a direct algebraic computation gives a lengthy but explicit formula for $\epsilon_g(q^*)$, as shown in [33].

**Approximate message passing, and its state evolution —** Our next result is based on an adaptation of a popular algorithm to solve random instances of generalized linear models, the AMP algorithm [15, 16], for the case of the committee machine and models described by (2).

The AMP algorithm can be obtained as a Taylor expansion of loopy belief-propagation (see [33]) and also originates in earlier statistical physics works [36, 37, 38, 39, 40, 26]. It is conjectured to perform the best among all polynomial algorithms in the framework of these models. It thus gives us a tool to evaluate both the intrinsic algorithmic hardness of the learning and the performance of existing algorithms with respect to the optimal one in this model.

The AMP algorithm is summarized by its pseudo-code in Algorithm 1, where the update functions $g_{\text{out}}, \partial_\omega g_{\text{out}}, f_W$ and $f_C$ are related, again, to the two auxiliary problems (3) and (4). The functions

**Algorithm 1** Approximate Message Passing for the committee machine

---

**Input:** vector $Y \in \mathbb{R}^m$ and matrix $X \in \mathbb{R}^{m \times n}$:

*Initialize*: $\hat{W}_i, g_{\text{out},\mu} \in \mathbb{R}^K$ and $\hat{C}_i, \partial_\omega g_{\text{out},\mu} \in \mathcal{S}_K^+$ for $1 \leq i \leq n$ and $1 \leq \mu \leq m$ at $t = 0$.

**repeat**

    Update of the mean $\omega_\mu \in \mathbb{R}^K$ and covariance $V_\mu \in \mathcal{S}_K^+$:

$$\omega_\mu^t = \sum_{i=1}^{n} \Big( \frac{X_{\mu i}}{\sqrt{n}} \hat{W}_i^t - \frac{X_{\mu i}^2}{n} \left(\Sigma_i^{t-1}\right)^{-1} \hat{C}_i^t \Sigma_i^{t-1} g_{\text{out},\mu}^{t-1} \Big) \quad | \quad V_\mu^t = \sum_{i=1}^{n} \frac{X_{\mu i}^2}{n} \hat{C}_i^t$$

    Update of $g_{\text{out},\mu} \in \mathbb{R}^K$ and $\partial_\omega g_{\text{out},\mu} \in \mathcal{S}_K^+$:

$$g_{\text{out},\mu}^t = g_{\text{out}}(\omega_\mu^t, Y_\mu, V_\mu^t) \quad | \quad \partial_\omega g_{\text{out},\mu}^t = \partial_\omega g_{\text{out}}(\omega_\mu^t, Y_\mu, V_\mu^t)$$

    Update of the mean $T_i \in \mathbb{R}^K$ and covariance $\Sigma_i \in \mathcal{S}_K^+$:

$$T_i^t = \Sigma_i^t \Big( \sum_{\mu=1}^{m} \frac{X_{\mu i}}{\sqrt{n}} g_{\text{out},\mu}^t - \frac{X_{\mu i}^2}{n} \partial_\omega g_{\text{out},\mu}^t \hat{W}_i^t \Big) \quad | \quad \Sigma_i^t = -\Big( \sum_{\mu=1}^{m} \frac{X_{\mu i}^2}{n} \partial_\omega g_{\text{out},\mu}^t \Big)^{-1}$$

    Update of the estimated marginals $\hat{W}_i \in \mathbb{R}^K$ and $\hat{C}_i \in \mathcal{S}_K^+$:

$$\hat{W}_i^{t+1} = f_W(\Sigma_i^t, T_i^t) \quad | \quad \hat{C}_i^{t+1} = f_C(\Sigma_i^t, T_i^t)$$

    $t = t + 1$

**until** Convergence on $\hat{W}, \hat{C}$.

**Output:** $\hat{W}$ and $\hat{C}$.

---

$f_W(\Sigma, T)$ and $f_C(\Sigma, T)$ are respectively the mean and variance under the posterior distribution (3) when $r \to \Sigma^{-1}$ and $Y_0 \to \Sigma^{1/2}T$, while $g_{\text{out}}(\omega_\mu, Y_\mu, V_\mu)$ is given by the product of $V_\mu^{-1/2}$ and the mean of $u$ under the posterior (4) using $\widetilde{Y}_0 \to Y_\mu$, $\rho - q \to V_\mu$ and $q^{1/2}V \to \omega_\mu$ (see [33] for more details). After convergence, $\hat{W}$ estimates the weights of the teacher-neural network. The label of a sample $X_{\text{new}}$ not seen in the training set is estimated by the AMP algorithm as

$$Y_{\text{new}}^t = \int \mathrm{d}y \Big( \prod_{l=1}^{K} \mathrm{d}z_l \Big) y P_{\text{out}}(y|\{z_l\}_{l=1}^K) \mathcal{N}(z; \omega_{\text{new}}^t, V_{\text{new}}^t), \tag{9}$$

where $\omega_{\text{new}}^t = \sum_{i=1}^n X_{\text{new},i} \hat{W}_i^t$ is the mean of the normally distributed variable $z \in \mathbb{R}^K$, and $V_{\text{new}}^t = \rho - q_{\text{AMP}}^t$ is the $K \times K$ covariance matrix (see below for the definition of $q_{\text{AMP}}^t$). We provide a demo of the algorithm on github [41].

AMP is particularly interesting because its performance can be tracked rigorously, again in the asymptotic limit when $n \to \infty$, via a procedure known as state evolution (a rigorous version of the cavity method in physics [14]), see [18]. State evolution tracks the value of the overlap between the hidden ground truth $W^*$ and the AMP estimate $\hat{W}_t$, defined as $q_{\text{AMP}}^t \equiv \lim_{n \to \infty} (\hat{W}^t)^\intercal W^*/n$, via:

$$q_{\text{AMP}}^{t+1} = 2 \frac{\partial \psi_{P_0}}{\partial r}(r_{\text{AMP}}^t), \qquad r_{\text{AMP}}^{t+1} = 2\alpha \frac{\partial \Psi_{P_{\text{out}}}}{\partial q}(q_{\text{AMP}}^t; \rho). \tag{10}$$

The fixed points of these equations correspond to the critical points of the replica free entropy (7).

Let us comment further on the convergence of the algorithm. In the large $n$ limit, and if the integrals are performed without errors, then the algorithm is guaranteed to converge. This is a consequence of the state evolution combined with the Bayes-optimal setting. In practice, of course, $n$ is finite and integrals are approximated. In that case convergence is not guaranteed, but is robustly achieved in all the cases presented in this paper. We also expect (by experience with the single layer case) that if the input-data matrix is not random (i.e. beyond our assumptions) then we will encounter convergence issues, which could be fixed by moving to some variant of the algorithm such as VAMP [42].

## 3 From two to more hidden neurons, and the specialization phase transition

**Two neurons —** Let us now discuss how the above results can be used to study the optimal learning in the simplest non-trivial case of a two-layers neural network with two hidden neurons, i.e. when model (1) is simply $Y_\mu = \text{sign}[\,\text{sign}(\sum_{i=1}^n X_{\mu i} W_{i1}^*) + \text{sign}(\sum_{i=1}^n X_{\mu i} W_{i2}^*)]$, with the convention that $\text{sign}(0) = 0$. We remind that the input-data matrix $X$ has i.i.d. $\mathcal{N}(0,1)$ entries, and the teacher-weights $W^*$ used to generate the labels $Y$ are taken i.i.d. from $P_0$.

In Fig. 1 we plot the optimal generalization error as a function of the sample complexity $\alpha = m/n$. In the left panel the weights are Gaussian (for both the teacher and the student), while in the center panel they are binary/Rademacher. The full line is obtained from the fixed point of the state evolution (SE) of the AMP algorithm (10), corresponding to the extremizer of the replica free entropy (7). The points are results of the AMP algorithm run till convergence averaged over 10 instances of size $n = 10^4$. In this case and with random initial conditions the AMP algorithm did converge in all our trials. As expected we observe excellent agreement between the SE and AMP.

In both left and center panels of Fig. 1 we observe the so-called *specialization* phase transition. Indeed (10) has two types of fixed points: A *non-specialized* fixed point where every element of the $K \times K$ order parameter $q$ is the same (so that both hidden neurons learn the same function) and a *specialized* fixed point where the diagonal elements of the order parameter are different from the non-diagonal ones. We checked for other types of fixed points for $K = 2$ (one where the two diagonal elements are not the same), but have not found any. In terms of weight-learning, this means for the non-specialized fixed point that the estimators for both $W_1$ and $W_2$ are the same, whereas in the specialized fixed point the estimators of the weights corresponding to the two hidden neurons are different, and that the network "figured out" that the data are better described by a non-linearly separable model. The specialized fixed point is associated with lower error than the non-specialized one (as one can see in Fig. 1). The existence of this phase transition was discussed in statistical physics literature on the committee machine, see e.g. [20, 23].

For Gaussian weights (Fig. 1 left), the specialization phase transition arises continuously at $\alpha_{\text{spec}}^G(K = 2) \simeq 2.04$. This means that for $\alpha < \alpha_{\text{spec}}^G(K = 2)$ the number of samples is too small, and the student-neural network is not able to learn that two different teacher-vectors $W_1$ and $W_2$ were used to generate the observed labels. For $\alpha > \alpha_{\text{spec}}^G(K = 2)$, however, it is able to distinguish the two different weight-vectors and the generalization error decreases fast to low values (see Fig. 1). For completeness we remind that in the case of $K = 1$ corresponding to single-layer neural network no such specialization transition exists. We show [33] that it is absent also in multi-layer neural networks as long as the activations remain linear. The non-linearity of the activation function is therefore an essential ingredient in order to observe a specialization phase transition.

The center part of Fig. 1 depicts the fixed point reached by the state evolution of AMP for the case of binary weights. We observe two phase transitions in the performance of AMP in this case: (a) the specialization phase transition at $\alpha_{\text{spec}}^B(K = 2) \simeq 1.58$, and for slightly larger sample complexity a transition towards *perfect generalization* (beyond which the generalization error is asymptotically zero) at $\alpha_{\text{perf}}^B(K = 2) \simeq 1.99$. The binary case with $K = 2$ differs from the Gaussian one in the fact that perfect generalization is achievable at finite $\alpha$. While the specialization transition is continuous here, the error has a discontinuity at the transition of perfect generalization. This discontinuity is associated with the 1st order phase transition (in the physics nomenclature), leading to a gap between algorithmic (AMP in our case) performance and information-theoretically optimal performance reachable by exponential algorithms. To quantify the optimal performance we need to evaluate the global extremum of the replica free entropy (not the local one reached by the state evolution). In doing so that we get that information theoretically there is a single discontinuous phase transition towards perfect generalization at $\alpha_{\text{IT}}^B(K = 2) \simeq 1.54$.

While the information-theoretic and specialization phase transitions were identified in the physics literature on the committee machine [20, 21, 3, 4], the gap between the information-theoretic performance and the performance of AMP —that is conjectured to be optimal among polynomial algorithms— was not yet discussed in the context of this model. Indeed, even its understanding in simpler models than those discussed here, such as the single layer case, is more recent [15, 26, 25].

**More is different —** It becomes more difficult to study the replica formula for larger values of $K$ as it involves (at least) $K$-dimensional integrals. Quite interestingly, it is possible to work out the solution of the replica formula in the large $K$ limit (thus taken *after* the large $n$ limit, so that $K/n$ vanishes). It is indeed natural to look for solutions of the replica formula, as suggested in [19], of the form $q = q_d I_{K \times K} + (q_a/K)\mathbf{1}_K \mathbf{1}_K^\mathsf{T}$, with the unit vector $\mathbf{1}_K = (1)_{l=1}^K$. Since both $q$ and $\rho$ are assumed to be positive, this scaling implies [33] that $0 \le q_d \le 1$ and $0 \le q_a + q_d \le 1$, as it should. We also detail in [33] the corresponding large $K$ expansion of the free entropy for the teacher-student scenario with Gaussian weights. Only the information-theoretically reachable generalization error was computed [19], thus we concentrated on the analysis of performance of AMP by tracking the state evolution equations. In doing so, we unveil a large computational gap.

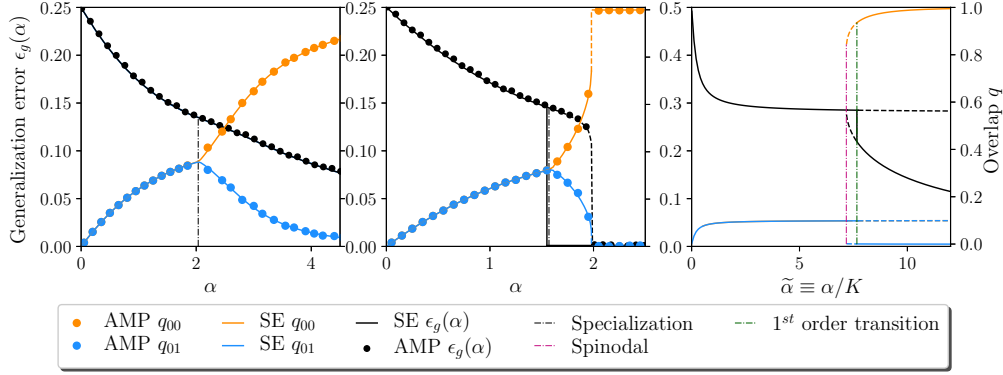

Figure 1: Order parameter and optimal generalization error for a committee machine with two hidden neurons with Gaussian weights (left), binary/Rademacher weights (center), and for Gaussian weights in the large number of hidden units limit (right). These are shown as a function of the ratio $\alpha = m/n$ between the number of samples $m$ and the dimensionality $n$. Lines are obtained from the state evolution equations (dominating solution is shown in full line), data-points from the AMP algorithm averaged over 10 instances of the problem of size $n = 10^4$. $q_{00}$ and $q_{01}$ denote diagonal and off-diagonal overlaps, and their values are given by the labels on the far-right of the figure.

In the right plot of Fig. 1 we show the fixed point values of the two overlaps $q_{00} = q_d + q_a/K$ and $q_{01} = q_a/K$ and the resulting generalization error. As discussed in [19] it can be written in a closed form as $\epsilon_g = \arccos\left[2\left(q_a + \arcsin q_d\right)/\pi\right]/\pi$. The specialization transition arises for $\alpha = \Theta(K)$ so we define $\widetilde{\alpha} \equiv \alpha/K$. The specialization is now a 1st order phase transition, meaning that the specialization fixed point first appears at $\widetilde{\alpha}^G_{\text{spinodal}} \simeq 7.17$ but the free entropy global extremizer remains the one of the non-specialized fixed point until $\widetilde{\alpha}^G_{\text{spec}} \simeq 7.65$. This has interesting implications for the optimal generalization error that gets towards a plateau of value $\varepsilon_{\text{plateau}} \simeq 0.28$ for $\widetilde{\alpha} < \widetilde{\alpha}^G_{\text{spec}}$ and then jumps discontinuously down to reach a decay aymptotically as $1.25/\widetilde{\alpha}$.

AMP is conjectured to be optimal among all polynomial algorithms (in the considered limit) and thus analyzing its state evolution sheds light on possible computational-to-statistical gaps that come hand in hand with 1st order phase transitions. In the regime of $\alpha = \Theta(K)$ for large $K$ the non-specialized fixed point is always stable implying that AMP will not be able to give a lower generalization error than $\varepsilon_{\text{plateau}}$. Analyzing the replica formula for large $K$ in more details [33], we concluded that AMP will not reach the optimal generalization for any $\alpha < \Theta(K^2)$. This implies a rather sizable gap between the performance that can be reached information-theoretically and the one reachable tractably. Such large computational gaps have been previously identified in a range of inference problems —most famously in the planted clique problem [27]— but the committee machine is the first model of a multi-layer neural network with realistic non-linearities (the parity machine is another example but use a very peculiar non-linearity) that presents such large gap.

## 4   Sketch of proof of Theorem 2.1

We denote $K$-dimensional column vectors by underlined letters. In particular $\underline{W}^*_i = (W^*_{il})^K_{l=1}$, $\underline{w}_i = (w_{il})^K_{l=1}$. For $\mu = 1, \ldots m$, let $\underline{V}_\mu, \underline{U}^*_\mu$ be $K$-dimensional vectors with i.i.d. $\mathcal{N}(0,1)$ components. Let $s_n \in (0, 1/2)$ a sequence that goes to 0 as $n$ increases, and let $\mathcal{M}$ be the compact subset of matrices in $S^{++}_K$ with eigenvalues in the interval $[1, 2]$. For all $M \in s_n\mathcal{M}$, $2s_n I_{K \times K} - M \in \mathcal{S}^+_K$. Let $\epsilon = (\epsilon_1, \epsilon_2) \in (s_n\mathcal{M})^2$. Let then $q : [0,1] \to \mathcal{S}^+_K(\rho)$ and $r : [0,1] \to \mathcal{S}^+_K$ be two "interpolation functions" (that will later on depend on $\epsilon$), and $R_1(t) \equiv \epsilon_1 + \int_0^t r(v)\mathrm{d}v$, $R_2(t) \equiv \epsilon_2 + \int_0^t q(v)\mathrm{d}v$. For $t \in [0,1]$, define the $K$-dimensional vector: $\underline{S}_{t,\mu} \equiv \sqrt{(1-t)/n} \sum_{i=1}^n X_{\mu i} \underline{W}^*_i + \sqrt{R_2(t)}\, \underline{V}_\mu + \sqrt{t\rho - R_2(t) + 2s_n I_{K \times K}}\, \underline{U}^*_\mu$ where matrix square-roots are well defined. We interpolate with auxiliary problems related to those discussed in sec. 2:

$$Y_{t,\mu} \sim P_{\text{out}}(\,\cdot\,|\,\underline{S}_{t,\mu}), \quad 1 \le \mu \le m, \qquad \underline{Y}'_{t,i} = \sqrt{R_1(t)}\,\underline{W}^*_i + \underline{Z}'_i, \quad 1 \le i \le n, \qquad (11)$$

where $\underline{Z}'_i$ is (for each $i$) a $K$-vector with i.i.d. $\mathcal{N}(0,1)$ components, and $\underline{Y}'_{t,i}$ is a $K$-vector as well. Recall that in our notation the $*$-variables have to be retrieved, while the other random variables are

assumed to be known (except for the noise variables obviously). Define now $\underline{s}_{t,\mu}$ by the expression of $\underline{S}_{t,\mu}$ but with $\underline{w}_i$ replacing $\underline{W}_i^*$ and $\underline{u}_\mu$ replacing $\underline{U}_\mu^*$. We introduce the *interpolating posterior*:

$$P_{t,\epsilon}(w,u|Y_t,Y_t',X,V) = \frac{1}{\mathcal{Z}_n(t,\epsilon)} \prod_i P_0(\underline{w}_i) e^{-\frac{1}{2}\|\underline{Y}_{t,i}' - \sqrt{R_1(t)}\underline{w}_i\|_2^2} \prod_\mu \frac{e^{-\frac{1}{2}\|\underline{u}_\mu\|_2^2}}{(2\pi)^{K/2}} P_{\text{out}}(Y_{t,\mu}|\underline{s}_{t,\mu})$$

where the normalization factor $\mathcal{Z}_n(t,\epsilon)$ equals the numerator integrated over all components of $w$ and $u$. The average free entropy at time $t$ is by definition $f_{n,\epsilon}(t) \equiv \mathbb{E}\ln\mathcal{Z}_n(t,\epsilon)/n$. One verifies, using in particular continuity and boundedness properties of $\psi_{P_0}$ and $\Psi_{P_{\text{out}}}$ (see [33] for details):

$$\begin{cases} f_{n,\epsilon}(0) & = & f_n - \frac{K}{2} + \mathcal{O}_n(1)\,, \\ f_{n,\epsilon}(1) & = & \psi_{P_0}(\int_0^1 r(t)\mathrm{d}t) + \alpha\Psi_{P_{\text{out}}}(\int_0^1 q(t)\mathrm{d}t;\rho) - \frac{1}{2}\mathrm{Tr}(\rho\int_0^1 r(v)\mathrm{d}v) - \frac{K}{2} + \mathcal{O}_n(1)\,. \end{cases}$$

Here $\mathcal{O}_n(1) \to 0$ in the $n,m\to\infty$ limit uniformly in $t$, $q$, $r$, $\epsilon$. The next step is to compute the free entropy variation along the interpolation path [33]:

**Proposition 4.1 (Free entropy variation)** *Denote by $\langle-\rangle_{n,t,\epsilon}$ the (Gibbs) expectation w.r.t. the posterior $P_{t,\epsilon}$. Set $u_y(x) \equiv \ln P_{\text{out}}(y|x)$. For all $t \in [0,1]$ the derivative $\frac{df_{n,\epsilon}(t)}{dt}$ equals*

$$-\frac{1}{2}\mathbb{E}\Big\langle \mathrm{Tr}\Big[\Big(\frac{1}{n}\sum_{\mu=1}^m \nabla u_{Y_{t,\mu}}(\underline{s}_{t,\mu})\nabla u_{Y_{t,\mu}}(\underline{S}_{t,\mu})^{\intercal} - r(t)\Big)\big(Q - q(t)\big)\Big]\Big\rangle_{n,t,\epsilon} + \frac{1}{2}\mathrm{Tr}\left[r(t)(q(t) - \rho)\right] + \mathcal{O}_n(1)\,,$$

*where $\nabla$ is the $K$-dimensional gradient w.r.t. the argument of $u_{Y_{t,\mu}}(\cdot)$, and $\mathcal{O}_n(1) \to 0$ in the $n,m\to\infty$ limit uniformly in $t$, $q$, $r$, $\epsilon$. Here, $Q_{ll'} \equiv \sum_{i=1}^n W_{il}^* w_{il'}/n$ is a $K\times K$ overlap matrix .*

A crucial step of the adaptive interpolation method is to show that the overlap concentrates (see [33]):

**Proposition 4.2 (Overlap concentration)** *Assume that for any $t \in (0,1)$ the transformation $\epsilon \in (s_n\mathcal{M})^2 \mapsto (R_1(t,\epsilon), R_2(t,\epsilon))$ is a $\mathcal{C}^1$ diffeomorphism with a Jacobian greater or equal to 1. Then one can find a sequence $s_n$ going to 0 slowly enough such that there exists a constant $C > 0$ and a $\gamma > 0$, that only depend on the support and moments of $P_0$ and on the activation $\varphi_{\text{out}}$ and $\alpha$, such that ($\|-\|_F$ is the Frobenius norm): $\mathrm{Vol}(s_n\mathcal{M})^{-2} \int_{(s_n\mathcal{M})^2} \mathrm{d}\epsilon \int_0^1 \mathrm{d}t\, \mathbb{E}\langle\|Q - \mathbb{E}\langle Q\rangle_{n,t,\epsilon}\|_F^2\rangle_{n,t,\epsilon} \le Cn^{-\gamma}$.*

Let $f_{\text{RS}}(q,r) \equiv \psi_{P_0}(r) + \alpha\Psi_{P_{\text{out}}}(q;\rho) - \mathrm{Tr}(rq)/2$ the *replica symmetric (RS) potential*. We have:

**Proposition 4.3 (Lower bound)** $\liminf_{n\to\infty} f_n \ge \sup_{r\in\mathcal{S}_K^+} \inf_{q\in\mathcal{S}_K^+(\rho)} f_{\text{RS}}(q,r)$.

*Proof:* Choose first $r(t) = r \in \mathcal{S}_K^+$ a fixed matrix. Then $R(t) = (R_1(t), R_2(t))$ can be fixed as the solution to the first order differential equation: $\partial_t R_1(t,\epsilon) = r$, $\partial_t R_2(t,\epsilon) = \mathbb{E}\langle Q\rangle_{n,t,\epsilon}$ and $R(0,\epsilon) = \epsilon$. We denote it $R(t,\epsilon) = (rt + \epsilon_1, \int_0^t q(v,\epsilon;r)\mathrm{d}v + \epsilon_2)$. It is possible to check (see [33]) that this ODE satisfies the hypotheses of the parametric Cauchy-Lipschitz theorem, and that by the Liouville formula the determinant of the Jacobian of $\epsilon \mapsto R(\epsilon,t)$ satisfies $J_{n,\epsilon}(t) = \exp\{\int_0^t \sum_{l\ge l'}^K \partial_{(R_2)_{ll'}}\mathbb{E}\langle Q_{ll'}\rangle_{n,s,\epsilon}(s,R(s,\epsilon))ds\} \ge 1$; indeed, this sum of partials is always positive, see [33]. Using then Prop. 4.1 and Prop. 4.2, we obtain $f_n = \mathrm{Vol}(s_n\mathcal{M})^{-2}\int_{(s_n\mathcal{M})^2} \mathrm{d}\epsilon f_{\text{RS}}(\int_0^1 q(v,\epsilon;r)\mathrm{d}v, r) + \mathcal{O}_n(1)$. This implies the lower bound.

**Proposition 4.4 (Upper bound)** $\limsup_{n\to\infty} f_n \le \sup_{r\in\mathcal{S}_K^+} \inf_{q\in\mathcal{S}_K^+(\rho)} f_{\text{RS}}(q,r)$.

*Proof :* We now fix $R(t) = (R_1(t), R_2(t))$ as the solution $R(t,\epsilon) = (\int_0^t r(v,\epsilon)\mathrm{d}v + \epsilon_1, \int_0^t q(v,\epsilon)\mathrm{d}v + \epsilon_2)$ to the following Cauchy problem: $\partial_t R_1(t,\epsilon) = 2\alpha\nabla\Psi_{P_{\text{out}}}(\mathbb{E}\langle Q\rangle_{n,t,\epsilon})$, $\partial_t R_2(t,\epsilon) = \mathbb{E}\langle Q\rangle_{n,t,\epsilon}$, and $R(0,\epsilon) = \epsilon$. We denote this equation as $\partial_t R(t,\epsilon) = F_n(R(t,\epsilon),t)$. It is then possible to verify that $F_n(R(t,\epsilon),t)$ is a bounded $\mathcal{C}^1$ function of $R(t,\epsilon)$, and thus a direct application of the Cauchy-Lipschitz theorem implies that $R(t,\epsilon)$ is a $\mathcal{C}^1$ function of $t$ and $\epsilon$ and by unicity of the solution, the function $\epsilon \mapsto R(t,\epsilon)$ is injective for any $t$. Since $\mathbb{E}\langle Q\rangle_{n,t,\epsilon}$ and $\rho - \mathbb{E}\langle Q\rangle_{n,t,\epsilon}$ are positive matrices (see [33]) we also have that $q(t,\epsilon) \in \mathcal{S}_K^+(\rho)$ and since by the differential equation we have $r(t,\epsilon) = 2\alpha\nabla\Psi_{P_{\text{out}}}(q(t,\epsilon))$ and as $\nabla\Psi_{P_{\text{out}}}(q) \in \mathcal{S}_K^+$ (see [33]), then $r(t,\epsilon) \in \mathcal{S}_K^+$. Moreover the Liouville formula for the Jacobian of the map $\epsilon \in (s_n\mathcal{M})^2 \mapsto R(t,\epsilon) \in R(t,(s_n\mathcal{M})^2)$

yields $J_{n,\epsilon}(t) = \exp\{\int_0^t \sum_{l \geq l'}^{K} [\partial_{(R_1)_{ll'}}(F_{n,1})_{ll'} + \partial_{(R_2)_{ll'}}(F_{n,2})_{ll'}](s, R(s,\epsilon))ds\}$. For all $s \in [0,1]$ the integrand $\sum_{l \geq l'}[\ldots] \geq 0$ (see [33]). We can again apply Prop. 4.2, and obtain

$$f_n = \frac{1}{\mathrm{Vol}(s_n \mathcal{M})^2} \int d\epsilon \{\psi_{P_0}(\int_0^1 r(\epsilon, v)dv) + \alpha \Psi_{P_{\mathrm{out}}}(\int_0^1 q(\epsilon, v)dv; \rho) - \frac{1}{2}\mathrm{Tr}\int_0^1 q(\epsilon, v)r(\epsilon, v)dv\} + \mathcal{O}_n(1).$$

By convexity of $\psi_{P_0}$ and $\Psi_{P_{\mathrm{out}}}$, $f_n \leq \frac{1}{\mathrm{Vol}(s_n \mathcal{M})^2} \int_{(s_n \mathcal{M})^2} d\epsilon \int_0^1 dv f_{\mathrm{RS}}(q(\epsilon, v), r(\epsilon, v)) + \mathcal{O}_n(1)$. We now remark that $f_{\mathrm{RS}}(q(\epsilon, v), r(\epsilon, v)) = \inf_{q \in \mathcal{S}_K^+(\rho)} f_{\mathrm{RS}}(q, r(\epsilon, v))$. Indeed, for every $r \in \mathcal{S}_K^+$, the function $g_r : q \in \mathcal{S}_K^+(\rho) \mapsto f_{\mathrm{RS}}(q, r) \in \mathbb{R}$ can be shown to be convex (see [33]) and its $q$-derivative is $\nabla g_r(q) = \alpha \nabla \Psi_{P_{\mathrm{out}}}(q) - r/2$. Since $\nabla g_{r(\epsilon,v)}(q(\epsilon, v)) = 0$ by definition of $r(\epsilon, v)$, and $\mathcal{S}_K^+(\rho)$ is convex, the minimum of $g_{r(\epsilon,v)}(q)$ is necessarily achieved at $q = q(\epsilon, v)$. Therefore

$$f_n \leq \frac{1}{\mathrm{Vol}(s_n \mathcal{M})^2} \int d\epsilon \int_0^1 dv \inf_{q \in \mathcal{S}_K^+(\rho)} f_{\mathrm{RS}}(q, r(\epsilon, v)) + \mathcal{O}_n(1) \leq \sup_{r \in \mathcal{S}_K^+} \inf_{q \in \mathcal{S}_K^+(\rho)} f_{\mathrm{RS}}(q, r) + \mathcal{O}_n(1),$$

which concludes the proof of Prop. 4.4. Combined with Prop. 4.3 we obtain Thm. 2.1.

## 5 Discussion

One of the contributions of this paper is the design of an AMP-type algorithm that is able to achieve the Bayes-optimal learning error in the limit of large dimensions for a range of parameters out of the so-called hard phase. The hard phase is associated with first order phase transitions appearing in the solution of the model. In the case of the committee machine with a large number of hidden neurons we identify a large hard phase in which learning is possible information-theoretically but not efficiently. In other problems where such a hard phase was identified, its study boosted the development of algorithms that are able to match the predicted threshold. We anticipate this will also be the same for the present model. We should, however, note that for larger $K > 2$ the present AMP algorithm includes higher-dimensional integrals that hamper the speed of the algorithm. Our current strategy to tackle this is to combine the large-$K$ expansion and use it in the algorithm. Detailed account of the corresponding results are left for future work.

We studied the Bayes-optimal setting where the student-network is the same as the teacher-network, for which the replica method can be readily applied. The method still applies when the number of hidden units in the student and teacher are different, while our proof does not generalize easily to this case. It is an interesting subject for future work to see how the hard phase evolves under over-parametrization and what is the interplay between the simplicity of the loss-landscape and the achievable generalization error. We conjecture that in the present model over-parametrization will not improve the generalization error achieved by AMP in the Bayes-optimal case.

Even though we focused in this paper on a two-layers neural network, the analysis and algorithm can be readily extended to a multi-layer setting, see [22], as long as the number of layers as well as the number of hidden neurons in each layer is held constant, and as long as one learns only weights of the first layer, for which the proof already applies. The numerical evaluation of the phase diagram would be more challenging than the cases presented in this paper as multiple integrals would appear in the corresponding formulas. In future works, we also plan to analyze the case where the weights of the second and subsequent layers (including the biases of the activation functions) are also learned. This could be done for instance with a combination of EM and AMP along the lines of [43, 44] where this is done for the simpler single layer case.

Concerning extensions of the present work, an important open case is the one where the number of samples per dimension $\alpha = \Theta(1)$ and also the size of the hidden layer per dimension $K/n = \Theta(1)$ as $n \to \infty$, while in this paper we treated the case $K = \Theta(1)$ and $n \to \infty$. This other scaling where $K/n = \Theta(1)$ is challenging even for the non-rigorous replica method.

## 6 Acknowledgements

This work is supported by the ERC under the European Union's Horizon 2020 Research and Innovation Program 714608-SMiLe, as well as by the French Agence Nationale de la Recherche under grant ANR-17-CE23-0023-01 PAIL. We gratefully acknowledge the support of NVIDIA Corporation with the donation of the Titan Xp GPU used for this research. Jean Barbier was supported by the Swiss National Foundation grant no 200021-156672. We also thank Léo Miolane for fruitful discussions.

## Footnotes

[2] The symbol $\mathbb{E}$ will generally denote an expectation over all random variables in the ensuing expression (here $\{X_{\mu i}, Y_\mu\}$). Subscripts will be used only when we take partial expectations or if there is an ambiguity.

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
