[Reviews · NeurIPS 2018]

Reviewer 1



The committee machine is a simple and natural model for a 2-layer neural network. (Here is the formal definition: it is a function R^n -> {+1,-1}, computed by taking K different linear combinations of the n inputs according to the weight matrix W (this is the hidden layer), then taking the sign of each hidden value, and then taking the majority vote of all these signs. The results of the paper also apply to many related models: you are allowed an arbitrary function mapping the K hidden values to the final binary output.) This paper studies the problem of learning the weights W under a natural random model. We are given m random examples (X,Y) where the input X (in R^n) is iid Gaussian and Y (in {+1,-1}) is the associated output of the network. The unknown weights W are iid from a known prior. The authors study the asymptotic regime where the dimension n goes to infinity with the ratio alpha = m/n held constant (recall m is the number of samples) and the number of hidden layers K held constant. Since the above is a fully-specified random model, it is amenable to study via tools from statistical physics. Prior work has rigorously analyzed the simpler case of "generalized linear models" which includes single-layer neural networks (K=1). Prior work has also analyzed the two-layer case using non-rigorous heuristics from statistical physics (which are well-established and widely believed to be reliable). The first contribution of the present work is to rigorously analyze the two-layer case (showing that the heuristics are indeed correct). Specifically, they give an exact formula for the limiting value (as n goes to infinity) of the information-theoretically optimal generalization error as a function of the sample complexity (alpha = m/n). The proof uses the "adaptive interpolation method", a recent method that has been successful for related problems. The second contribution of the present work is to give a polynomial-time algorithm for learning the weights W based on AMP (approximate message passing). They also give an exact formula for the generalization error (as n goes to infinity) that it achieves. AMP is a well-established framework and is widely conjectured to be optimal among all polynomial-time algorithms for this type of problem. By comparing the formulas for AMP's performance and the information-theoretically optimal performance, the authors identify regimes that appear to have inherent statistical-to-computational gaps. The authors give a thorough investigation of these gaps and of the various "phase transitions" occurring in the problem. I think this is a strong paper that builds on a lot of deep and powerful machinery from statistical physics in order to give important and fundamental advances towards a rigorous understanding of neural networks. The paper is well-written. I would be interested if the authors could comment on whether there is any hope of extending the results to more than two layers. EDIT: I have read the other reviews and the author feedback. My opinion of the paper remains the same -- I vote to accept it.

Reviewer 2



This paper has two contributions to the committee machine, which is a two-layered neural network with a fixed hidden-to-output connectivity and trainable input-to-hidden connectivity. First, Author(s) proposed a rigorous proof of the replica formula by using the adaptive interpolation method. Second, they derived the AMP algorithm for this model and mainly argued the computational gap, which implies that it is hard for polynomial algorithms to reach the information-theoretical performance. The rigorous proof part (Section 2 &4) seems to be a bit too technical, and it might be hard to collect a great deal of interests. However, this types of theory would be useful to more develop the statistical mechanical frameworks to analyze learning systems. To collect more general interests, I think that it would be better for Author(s) to give comments on the following things. * How trivial is it to generalize the committee machines to ones with bias parameters (that is, local magnetic fields in statistical physics)? In particular, I am wondering whether the phase transition suffers from any qualitative change. *Recently, studies in machine learning have intensively analyzed the existence of spurious local minima in shallow networks. For instance, in shallow ReLU networks, if the teacher and student networks have the same number of hidden units, there exist spurious minima [Safran&Shamir ICML2018]. In contrast, the number of hidden units of the teacher net are less than that of the student net, the spurious minima empirically disappear. Other related works are also mentioned in this paper. I think it would be better for Author(s) to mention this topic and to relate it to your results. In particular, I am wondering whether the computational gap occurs because the AMP gets stuck in the local spurious solutions or not. * Does the AMP (algorithm 1) always converge to a unique solution? It would be better to mention the suboptimality or possible explosion of the algorithm if they might happen. Typo (probably): in Line 188, lower error that -> lower error than

Reviewer 3



The authors provide a rigorous justification of the statistical physics approaches for studying the learning and generalization capabilities of shallow NN dealing with random data, in the so called teacher-student setting. They also analyze the behaviour of an approximate message passing (AMP) algorithm. Interestingly enough, they discuss a regime in which AMP fails and yet a low generalization error is information-theoretically achievable. The existence of this computational obstacle was known; however providing a rigorous setting is important, given that the statistical physics tools appear to be among the most powerful ones to analyze learning problems.